# Precision in 3D: A Fast and Accurate Algorithm for Reproducible Motoneuron Structure and Protein Expression Analysis

**DOI:** 10.3390/bioengineering12070761

**Published:** 2025-07-14

**Authors:** Morgan Highlander, Shelby Ward, Bradley LeHoty, Teresa Garrett, Sherif Elbasiouny

**Affiliations:** 1Department of Biomedical, Industrial, and Human Factors Engineering, College of Engineering and Computer Science, Wright State University, Dayton, OH 45435, USA; 2Boonshoft School of Medicine, Wright State University, Dayton, OH 45435, USA; 3Department of Neuroscience, Cell Biology and Physiology, College of Science and Mathematics, Wright State University, Dayton, OH 45435, USA

**Keywords:** motoneuron, volumetric analysis, structural analysis, protein expression

## Abstract

Structural analysis of motoneuron somas and their associated proteins via immunohistochemistry (IHC) remains tedious and subjective, requiring costly software or adapted 2D manual methods that lack reproducibility and analytical rigor. Yet, neurodegenerative disease and aging research demands precise structural comparisons to elucidate mechanisms driving neuronal degeneration. To address this need, we developed a novel algorithm that automates repetitive and subjective IHC analysis tasks, enabling thorough, objective, blinded, order-agnostic, and reproducible 3D batch analysis. With no manual tracing, the algorithm produces 3D Cartesian reconstructions of motoneuron somas from 60× IHC images of mouse lumbar spinal tissue. From these reconstructions, it measures 3D soma volume and efficiently quantitates net somatic protein expression and macro-cluster size. In this validation study, we applied the algorithm to assess soma size and C-bouton expression in various healthy control mice, comparing its measurements against manual measurements and across multiple algorithm users to confirm its accuracy and reproducibility. This novel, customizable tool enables efficient and high-fidelity 3D motoneuron analysis, replacing tedious, qualitative, cell-by-cell manual tuning with automatic threshold adaptation and quantified batch settings. For the first time, we attain reproducible results with quantifiable accuracy, exhaustive sampling, and a high degree of objectivity.

## 1. Introduction

Analysis of immunohistochemistry (IHC) images for measuring cell structures and proteins is a tedious, manual, and subjective process. Existing commercial software programs for analysis (such as Neurolucida and Imaris) are very expensive and primarily developed for nuances of analysis in brain tissue. These platforms are difficult to use for isolating somas and analyzing protein expression of motoneurons in the spinal cord, where protein structures are more varied in size and somas are much larger and less-densely packed. Spinal motoneuron protein analysis with existing software tools requires manual cell-by-cell threshold adjustments and cluster-size tuning based on subjective visual interpretations of the cell bodies and protein structures. These manual adjustments are qualitative, subject to bias, and largely untraceable, contributing to lack of transparency and lack of reproducibility of resulting measurements. Additionally, as the number of cells analyzed increases, so does the heavy manual analyzer burden and the difficulty for maintaining consistency in the subjective tuning. As a result, researchers have resorted to custom 2D analysis methods [1,2,3,4,5,6] to ease the analysis burden, more tightly control the approach, and increase the number of cells analyzed. However, these approaches remain time-consuming and inefficient, plagued by similar subjectivity and additional stereological cluster sampling limitations that hinder reproducibility [2] and reduce sensitivity to detecting biological changes. Critically, small methodological decisions in these analyses can drastically alter study outcomes—even reversing conclusions [2]. Despite these challenges, obtaining accurate and reproducible measurements is essential for detecting subtle changes in protein expression, which may help uncover previously unknown mechanisms driving neuronal vulnerability, aging, and degeneration.

To meet the demands for analytical efficiency, specificity, objectivity, and reproducibility, we developed an automated pipeline that streamlines key components of IHC image analysis, with an emphasis on spinal tissue analysis. Our algorithm generates 3D reconstructions of motoneuron somas and quantifies 100% of the protein labeling on the somatic membrane. Using Canny edge detection and a custom Cartesian reconstruction approach, the algorithm produces accurate 3D soma reconstructions, enabling highly precise and consistent soma size measurements. These reconstructions also define the precise 3D image coordinates for analyzing total somatic protein expression and for automatically identifying and quantifying all membrane-associated protein clusters. All cluster analysis settings are quantified and automatically tracked, with built-in setting sensitivity analysis and automatic threshold adjustments per cell. In this validation study, we present a series of performance assessments and comparisons to existing methods to evaluate the accuracy and robustness of our approach. Due to the absence of definitive ground-truth labels for protein clusters in these noisy biological images, we prioritized a validation strategy centered on geometric benchmarking, cross-condition consistency, robustness to parameter (mainly threshold) shifts, and reproducibility across users over empirical accuracy metrics for image segmentation. This strategy directly reflects the operational goals of the algorithm in its intended biological context. Although the algorithm is adaptable for analyzing any membrane-associated protein, we used the C-bouton—an example of a macro-clustering synaptic protein—as the primary case study for algorithm validation.

## 2. Materials and Methods

### 2.1. Three-Dimensional Cartesian Reconstructions

We used Olympus FV1000 (.oib) 60× IHC (Olympus Corporation, Tokyo, Japan) confocal images of quadruple-labeled mouse lumbar spinal tissue collected for various study controls over the last decade in our lab. In the example macro-analysis for this validation study, we used Nissl (NeuroTrace, Life Technologies, cat #: N21479, Thermo Fisher Scientific, Carlsbad, CA, USA) or NeuN (Millipore-Sigma, cat#ABN90) as our soma label, and VAChT (Millipore-Sigma, cat #: ABN100, Sigma Aldrich, St. Louis, MO, USA) or ChAT (Novus Biologicals, cat #: NBP246620, Toronto, ON, Canada) as our alpha motoneuron-specific indicator and label for our protein of interest, the C-bouton.

Figure 1 shows the overall flow of the algorithm from the data inputs and user inputs (shown in the dashed boxes) to the soma and cluster measurement outputs. First, the algorithm converts the .oib or .oir Olympus images to standard .tif frames for each of the 4 labels. Then, the user loads the soma body label frames into our GUI, which uses a triangle-algorithm [7] threshold approach with custom-built Canny edge-detection to automatically outline the somas. The user scrolls through the frames to identify cells of interest and then queues the algorithm to batch-process the reconstructions. The algorithm automatically finds the outline of the identified motoneuron somas through the stack of images (Figure 2C) and generates 3D region-of-interest (ROI) Cartesian matrices indicating the location of the soma edge. These Cartesian matrices are loaded into our visualizer and displayed as clean and minimalist 3D soma reconstructions (blue 3D soma with colored protein clusters in Figure 2D), a marked improvement over the clunky and background-cluttered visualizations offered by current software like Neurolucida (Figure 2B). The reconstructions are then converted to a list of X,Y,Z coordinates, defining each soma membrane and thus enabling the algorithm to perform automatic soma size and somatic protein expression analysis in large batches, with minimal human oversight required.

### 2.2. Somatic Morphology Measurements

Traditionally, spinal motoneuron soma size has been estimated by scrolling through stacks of 2D image frames and manually tracing the visually identified largest cross-sectional area (LCA) of the somas (see Figure 2A). Similarly, but with more accuracy and consistency, our algorithm counts the pixels contained inside the automatic traces of our 3D Cartesian reconstructions and returns the true LCA in the z plane in μm^2^. Additionally, our algorithm measures the 3D volume in μm^3^ and surface area in μm^2^ of the soma reconstructions, using similar voxel (3D pixel) counting.

### 2.3. Protein Expression Analysis

The XYZ coordinates of the 3D soma reconstruction are used to create a “shell” of the soma membrane that includes pixels within a given membrane search radius (MSR) from the defined 3D soma edge (default is 2 μm). This search radius must be small enough to avoid protein expression on neighboring structures, yet large enough to capture the entire volume of expression on the soma membrane. Once the membrane shell is defined, the .tif frames for the labeled protein of interest are then loaded into MATLAB (R2023A), and the 4-dimensional (X,Y,Z, intensity) data is extracted for coordinates in the soma membrane shell. Next, the protein intensity values are used to calculate a label threshold. Pixels above the calculated threshold intensity are considered labeled for that protein, and their spatial distributions are analyzed to identify distinct protein macro-clusters with our custom-built density-based spatial-clustering application with noise (DBSCAN)-clustering algorithm.

Specifically, labeled pixels (those brighter than the threshold) with enough labeled neighbors are assigned to a cluster using our DBSCAN-clustering algorithm, and then clumps of clusters are combined where appropriate. DBSCAN is a density-based approach for clustering nearby points that does not require a predetermined number of clusters, which is ideal for our application, where the number of clusters is initially unknown. Following cluster identification, the algorithm then generates a simplified 3D visualization of the clusters on the soma membrane (Figure 2D) and measures each cluster’s LCA and volume. Further, the algorithm calculates 1-net expression of the protein clusters per cell as the total number of clusters and total volume of clusters, and 2-relative net expression as the cluster density (# of clusters per membrane size) and the relative total cluster volume (total cluster volume per membrane size).

### 2.4. Validation: Reproducibility and Sensitivity

Soma size measurements were validated (see Figure 3) with quantified accuracy using known shapes of three different sizes (small, medium, and large), with different levels of curvature: cube (no curvature), sphere (symmetric curvature), and ellipsoid (asymmetric curvature). The three sizes were selected to encompass the range of physiological motoneuron soma sizes. The algorithm-measured soma LCA, surface area, and volume were compared to the mathematically calculated sizes (Figure 3). Further, we compared the algorithm’s sampling to manual methods by measuring the % of membrane analyzed for cluster density and the number of clusters measured for size (Figure 2E). We also compared algorithm measurements directly to standard manual measurements on the same cells: soma LCA, cluster LCA, and cluster density (Figure 4 and Figure 5). The reproducibility of our new algorithmic approach for soma measurements was tested by comparing results of 2 users who independently reconstructed and measured somas on the same set of cells from various control mice at different ages (Figure 6). We visually verified our protein labeling We likewise tested the reproducibility of our algorithm’s cluster measurements by comparing the results of 2 independent users (Figure 7). However, we eliminated external sources of variability in our cluster comparisons to strengthen our validation and determine the most appropriate protein cluster measurements. We narrowed our subject pool to 3 control animals from the same animal line, age, and sex, collected in a single randomized and batch-controlled study with highly consistent antibody batches and labeling methodology. Then, we characterized how our algorithm captures biological variability by comparing the measured protein expression between the 3 control animals (Figure 7).

### 2.5. Thresholding

Thresholding is a vital part of our analysis, as it could greatly impact the measurements of the algorithm. Complicating the issue, image background and off-target binding artifact can be highly variable due to the nature of IHC labeling. Nonetheless, it is important that the chosen threshold represents protein expression accurately and can fairly represent label across study groups with a variety of image-background conditions inherent to the IHC process. We initially picked our thresholding method by visually minimizing background and maximizing visually identified “true” protein label during our algorithm-tuning process for the C-bouton macro-clusters. We used a randomly selected subset of cells from each study group for this tuning. Our chosen thresholding method is a custom implementation of the triangle algorithm [7], with a default shift of 10%. We applied this thresholding algorithm on the membrane shell separately for each 2D image frame, since we found that label intensity and background vary most in the z-direction due to the normal gradient of antibody penetration. Though we were visually satisfied with our chosen threshold shift of 10%, we wanted to objectively determine if the threshold level would impact our findings. Thus, we performed a threshold-sensitivity analysis to explore how each output parameter of the algorithm is changed over a large range of threshold shifts (see Figure 8).

Furthermore, to simplify the thresholding process for future studies implementing our algorithm, we used our threshold-sensitivity results and our extensive and methodical manual observations of the visualized clusters at different thresholds to develop threshold stability-range criteria. The stability range was then used to automatically calculate an optimum threshold-shift suggestion per cell to maximize the stability, reasonability, and reproducibility of our protein cluster measures (see Figure 9).

### 2.6. Statistics

All data was analyzed in MATLAB as follows. All groups were tested for normality using Shapiro–Wilk’s test (swtest). All groups were tested for equal variance with their comparison groups, using Bartlett’s equal variance test (vartestn). Parameters with no groups that violated normality and with equal variance between groups were analyzed by ANOVA (anovan) and post hoc Tukey’s for ANOVA (multcompare). Most parameters violated either normality in at least one group or equal variance between groups, and so they were compared by the Kruskal–Wallis non-parametric test (kruskalwallis) with post hoc Tukey’s for Kruskal–Wallis (multcompare). Since the non-parametric tests were used most often, all data is reported as box plots with median line, with boxes indicating the inner-quartile range, and whiskers indicating the range of non-outliers. Correlation coefficients (ρ) were Pearson’s (corrcoef). Paired tests were performed with Wilcoxon signed-rank test (signrank) since non-normality was indicated. All tests were performed with an alpha of 0.05. One asterisk indicates *p* < 0.05, two asterisks indicate *p* < 0.01, and three asterisks indicate *p* < 0.001.

## 3. Results

### 3.1. Somatic Morphology Measurements

#### 3.1.1. Algorithm Performance on Known Geometries

To evaluate the algorithm’s performance, we first tested it on a variety of geometrical objects (cubes, spheres, and ellipsoids) with known areas and volumes that fall within the physiological range of spinal motoneurons. We knew there would be some level of measurement error since images comprise 3D pixels, or voxels, that have straight edges and therefore cannot perfectly represent smooth, curved surfaces, as illustrated in the bottom-center panel of Figure 3. The goal was to quantify this error at our sampling resolutions. In Figure 3, our testing captures expected performance variance depending on curvature level (straight-edged objects will have less error since voxels are also straight-edged) and dimension resolution (2D will have one less dimension of curvature–voxel error than 3D). Despite these factors, we see remarkable consistency in the algorithm’s volume measurements.

With regard to curvature, algorithm measurements are most accurate for non-curved geometries because images are composed of straight-edged 3D pixels (voxels). Consequently, the algorithm has the highest accuracy with cubes for all size parameters (Figure 3, the three cubes have accuracy close to 100%). Additionally, due to dimension resolution, both 3D measures (volume and surface area, Figure 3 bottom 2 rows) exhibit lower accuracy than 2D largest cross-sectional area (LCA, Figure 3 top row) due to their additional, lower resolution z-dimension (0.3 μm/z-plane compared to 0.172 μm/xy-pixel). While resolutions can vary between studies, it is standard for the z-plane to have lower resolution than the xy-plane, primarily due to the greater time and computational demands associated with higher-resolution z-steps (i.e., capturing more images per tissue volume analyzed).

However, despite its higher measurement accuracy, LCA remains a less specific indicator of cell size due to its 2D nature, which ignores the depth of the soma and is biased by the orientation of the soma relative to the image plane. Therefore, it is important to establish algorithm performance on the 3D size measures, which do account for soma depth and orientation. Our volume measurements show a remarkably consistent underestimation of ~23%, regardless of the size or orientation of the curved shapes (see orange box in Figure 3). This systematic underestimation enables correction, allowing us to establish refined soma size criteria for motoneuron classification. Previously, an LCA threshold of 300 μm [2] was used to identify motoneurons [8]. Assuming a spherical soma, this LCA corresponds to a volume of ~3900 μm^3^, which—after accounting for the ~23% underestimation—would be measured at ~3000 μm^3^. Accordingly, based on our volumetric data, we have revised the classification criteria for alpha motoneurons to either an LCA of ≥300 μm^2^ or a volume of ≥3000 μm^3^ for more accurate identification.

Unlike volume, surface area is unreliable for estimating the size of non-square objects from images. Our validation (see Figure 3, middle row) illustrates this unreliability, demonstrating inconsistent and unpredictable surface area estimates that jump between over- and underestimation.

Accordingly, our results suggest that volume represents a more reliable metric for assessing 3D soma size. To achieve physiological accuracy, a 23% volume correction factor should be applied to compensate for the anticipated underestimation discussed above. In summary, Figure 3 validates the accuracy of our algorithmic soma reconstruction measurements, reinforcing that volume is a more robust and consistent metric than surface area for estimating the 3D size of highly curved, asymmetrical, and randomly oriented objects such as motoneuron somas.

#### 3.1.2. Comparisons of Manual-to-Algorithm Soma LCA

After validating the algorithm’s performance and quantitating the accuracy of our soma measurements on geometrical objects, we proceeded to compare the algorithm’s measurements to manual measurements of real cells. In Figure 4, we reconstructed and measured somas that had been previously analyzed manually and evaluated the correlation coefficient (ρ) between the algorithm’s measurements and the manually collected data. Our analysis revealed a strong correlation (ρ = 0.92) between the two methods, indicating that the algorithm reliably measures soma LCAs similarly to manual tracing (Figure 4A, scatterplot), albeit slightly smaller (Figure 4A, boxplot, *p* < 0.001). Notably, algorithm performance tests in Figure 3 demonstrated high accuracy in LCA measurements, with minor algorithm overestimations (see purple highlights in Figure 3, top row) compared to true (calculated) LCA. However, manual measurements exceeded even the algorithm’s slight overestimations, indicating a more pronounced overestimation error in manual traces. This discrepancy is likely due to the inherent subjectivity of manual tracing and the common practice of increasing cell body label brightness during tracing for better visibility, which can artificially expand the perceived soma boundary. Taken together, these findings confirm that the algorithm measures soma LCAs more precisely and with less overestimation than manual methods.

#### 3.1.3. Reproducibility and Specificity of 2D and 3D Soma Size Measures

After verifying the accuracy of the algorithm’s soma size measurements by comparing its output to manual measurements, the next step was to assess the reproducibility of these measurements when comparing soma sizes across different groups, even when different individuals analyzed the same cells.

Reproducibility: To evaluate the reproducibility of the algorithm’s soma size measurements (Figure 6), two independent users (user 1 and user 2) employed the algorithm to measure motoneuron size across three different age groups: pup (P7–P9, N = 3 mice, n = 12 cells), adolescent (P40–42, N= 3 mice, n = 34 cells), and young adult (P90, N = 4 mice, n = 22 cells). Both users independently reconstructed and analyzed the same cells using the algorithm. Given the natural course of development, we hypothesized that motoneuron somas in older groups would be larger than those in the pup group.

Both the 2D LCA measurements (Figure 6A) and 3D volume measurements (Figure 6B) detected the smaller soma size in pups and produced consistent and reproducible results across users, as evidenced by the identical statistical difference indicators in the left and right graphs. This consistency supports the robustness of the algorithm for comparative analyses of soma sizes. In summary, we determined that the algorithm is highly reproducible across users of the algorithm for comparisons of soma size using LCA or volume.

Specificity: Given that both 2D LCA and 3D volume measurements were shown to be reproducible with the algorithm, the next step was to evaluate whether 3D volume provided any distinct advantages over the traditional 2D LCA measure, or if LCA alone was sufficient for assessing soma size. A comparison of the changes observed in Figure 6A (2D LCA) and 6B (3D volume) reveals that while both methods successfully detected the expected size difference—showing adolescent somas as larger than pup somas—only the 3D volume measurement was sensitive enough to distinguish that young-adult somas were also larger than pup somas. This additional differentiation captured by 3D volume, but missed by 2D LCA, underscores volume’s higher specificity in measuring soma size. By incorporating the 3rd dimension, 3D volume provides a more precise and sensitive representation of soma size, enabling it to detect subtle differences that 2D LCA might overlook.

Our findings show that the algorithm reliably detects soma size changes across users, with 3D volume emerging as the most specific measure of soma size. In contrast, while 2D LCA is reproducible, it may not capture finer size variations. Overall, from Figure 6, we conclude that 3D volume is the most reproducible and specific measure for soma size, while 2D LCA remains reproducible but less sensitive to subtle size differences.

### 3.2. Macro-Cluster Analysis

The previous sections demonstrated the algorithm’s effectiveness in analyzing neuron soma size properties. In this section, we expand its application to evaluate motoneuronal protein expression levels, with a particular focus on C-bouton expression, showcasing the algorithm’s broader utility.

#### 3.2.1. Comparisons of Manual-to-Algorithm Macro-Cluster LCA and Density (Figure 4B–D)

To evaluate novel components of the algorithm, like automatic cluster detection and label brightness thresholding, we sought to compare the algorithm’s cluster measurements directly to manual measures. This comparison gives perspective to how the new approach represents the clusters compared to old manual methods. Namely, we focused on the cluster LCA and density, since we can measure them manually for direct comparison.

Cluster LCA: Based on prior observations with soma LCA measurements (Figure 4A), we anticipated that algorithm-derived cluster LCAs would also be smaller than those obtained manually. In Figure 4B, we compared manual and algorithm-generated cluster LCAs from 12 cells, using a sampling strategy that included 5–8 en-fosse clusters per cell [2]. To enhance the robustness of these comparisons, we maximized biological variability by randomly selecting one cell each from 12 mice—specifically, 4 young-adult, 4 middle-aged, and 4 old mice, comprising 2 males and 2 females per age group. We analyzed a total of 79 cluster LCAs manually (limited to en-fosse clusters with substantial surface area in the 2D image plane) and compared the algorithm’s LCA measurements of the same 79 clusters across the 12 cells (Figure 4B; also see Figure 5 for visual cluster comparisons).

We found a positive correlation between algorithm and manual LCAs (Figure 4B, scatterplot; Pearson’s ρ = 0.57). However, the algorithm-measured LCAs were smaller than manual measurements (Figure 4B, boxplot, *p* < 0.001, Wilcoxon paired sign-rank test). This discrepancy is likely due to several factors, which can be visualized in Figure 5. (1) Elliptical approximation in manual tracing: Manual traces tend to approximate cluster boundaries as elliptical (see Figure 5A), which is less precise than the pixel-level edge detection used by the algorithm (see Figure 5B). (2) Inaccurate filling assumptions: Manual methods assume that all pixels within the traced boundary are part of the cluster, despite the reality that protein distributions are not perfectly contiguous. These missing areas within the boundaries of the cluster edge are counted toward the manual-cluster LCA, but not the algorithm-cluster LCA. (3) Visualization bias: Manual tracing often requires increasing label brightness to enhance visualization and ensure that enough en-fosse clusters are identified (5–8 per cell). This variability in brightness adjustments inflates the visual area of the protein label in manual analysis, leading to inconsistencies in cluster size estimation.

To further validate the algorithm’s accuracy, we performed extensive 2D visualizations of algorithm-identified clusters overlayed on the original raw image intensities, as illustrated in Figure 5. This 1:1 visual confirmation consistently supported the appropriateness and precision of the algorithm’s cluster identifications.

Together, the analysis shows that while algorithm-measured cluster LCAs are smaller than manual measurements, they are more precise and less subjective, minimizing errors introduced by visualization adjustments and assumptions inherent to manual tracing. This evidence underscores the reliability of the algorithm for accurate cluster size analysis.

Cluster density: In addition to the 1:1 cluster LCA comparisons, we performed 1:1 cell comparisons of the algorithm’s cluster density measurements against manual density estimations. Manual methods for measuring cluster density typically involve visually counting the number of clusters within a defined region of the soma and normalizing this count to the region’s size. In this study, we applied a standard approach [9] where the average number of clusters was counted on a 2D plane positioned at three levels relative to the LCA: 2 microns above, at the LCA, and 2 microns below. This average cluster count was then normalized by the LCA perimeter to estimate density as follows:(1)$$\mathrm{manual}\;\mathrm{density}\;\mathrm{approximation}=\frac{(n_{above}+n_{LCA}+n_{below})/3}{LCAperimeter}\;\;\;\;\;\;\;\;\;\;\;\;\;\;\;(\#/\mu m),$$

This manual method examines clusters in only about 25% of the soma membrane (represented by the red band in Figure 2D) and extrapolates this sample to estimate cluster density for the entire soma. In contrast, the algorithm allows for a more comprehensive and accurate measurement by automatically counting clusters across 100% of the soma membrane. The algorithm then calculates true cluster density by dividing the total number of clusters by the membrane size as follows:(2)$$\mathrm{algorithm}\;3D\;\mathrm{density}=\frac{n_{all}}{membrane\;size}\;\;\;\;\;\;\;\;\;\;\;\;\;\;\;(\#/{\mu m}^{3}),$$

Membrane size is defined as the volume of the image region that was searched for clusters, effectively representing the soma membrane surface area with a thickness corresponding to the algorithm’s search radius. Due to differences in methodologies and units—manual density estimations are reported as clusters per µm of perimeter, whereas algorithm density measures are expressed as clusters per µm^3^ of membrane volume—a direct comparison of their absolute values is not feasible. Instead, we aimed, in this section, to determine which method is more reliable and reproducible.

To assess this, we selected a subset of 20 cells (2 cells from each animal for three adolescent, four young-adult, and three middle-aged animals) and performed independent density measurements using both methods. Specifically, two users (user 1 and user 2) trained in manual analysis independently measured C-bouton density for these 20 cells. Likewise, two separate users (user 1 and user 2) trained in algorithm-based analysis independently reconstructed the same 20 cells and measured cluster density using the algorithm.

In Figure 4C, we compare the correlations between manual and algorithm-based density measurements for the four users (manual user 1, manual user 2, algorithm user 1, and algorithm user 2). The color distinction separates the manual users, while the shade distinction separates the algorithm users. Specifically, comparisons with manual user 1 are shown in pink, while comparisons with manual user 2 are shown in blue. Darker shades indicate comparisons to algorithm user 1, while lighter shades indicate comparisons to algorithm user 2.

The results show significant positive correlations across all user combinations (see the reported statistics in Figure 4C), but notably, the linear regression lines are grouped more distinctly by color (manual user) than by shade (algorithm user). This clustering by color suggests that manual user variability had a greater impact on the measurements than algorithm user variability.

To further explore this observation, we performed a pairwise user–user comparison of the manual density measurements for each cell. The results revealed an alarmingly high percent difference of 51.2% between the two manual users (Figure 4D). This variability resulted from inconsistencies in user approaches: one user struggled to identify clusters on some cells, while the other tended to adjust label intensities more aggressively, resulting in the identification of more clusters. We retained this variability in our study to reflect the significant influence that small procedural differences can have on manual measurements, even among users who received identical training.

We then performed a pairwise comparison of algorithm-based density measurements from the two separate users. The results demonstrated a substantial reduction in variability: the average difference between algorithm users was just 16.4% (see Figure 4D, boxplots), representing a 2/3 reduction compared to 51.2% difference observed in manual methods (Figure 4D, boxplots).

Altogether, the results indicate that although algorithm-based cluster density measurements are significantly correlated with manual estimations across various users, the algorithm markedly reduces user-dependent variability. This enhanced consistency highlights the algorithm’s superior reliability and reproducibility for measuring cluster density.

#### 3.2.2. Reproducibility of Macro-Cluster Analysis

As previously documented [2] and illustrated in our results so far, there is a lack of reproducibility in manual cluster measurements, leading to conflicting results across macro-cluster comparisons. To address this, we aimed to confirm the reproducibility of the algorithm’s macro-cluster measurement comparisons across multiple users (user 1 and user 2). In Figure 7, user 1 and user 2 independently compared C-bouton expression between animals on the same cohort of 60 cells from three 3-month-old male mice, using the algorithm alongside their separate soma reconstructions. Given that prior methods have struggled to produce consistent, reproducible measurements of C-bouton expression [2], we were uncertain which parameters would align or differ between control animals. Nevertheless, we hypothesized that cluster measurements should be highly consistent among these animals, as they were processed under tightly controlled conditions with a single batch of antibodies and identical protocols. Using the algorithm, we measured over 12 times the number of clusters per cell compared to prior methods (see Figure 2E) and verified reproducibility by comparing independently obtained results from both users (see Figure 7).

In Figure 7A,B, both users identified statistically significant differences in net C-bouton expression, with animal A exhibiting lower net C-bouton expression (cluster count and total cluster volume) than animals B and C. Similarly, in Figure 7C, both users observed reduced C-bouton density in animal A compared to animals B and C. Like density, total relative cluster volume is also indicative of relative net expression, as it is also measured relative to soma membrane size (see Figure 1). However, unlike density, relative total cluster volume did not significantly differ among the control animals, despite trending similarly (user 1: *p* = 0.077; user 2: *p* = 0.050). Importantly, both users measured smaller soma membranes for animal A compared to animals B and C (Figure 7G). This suggests that the observed biological variability in net C-bouton expression (Figure 7A,B) may, in part, be influenced by the size of the cells themselves. Furthermore, the changes in density and the trends in relative total volume imply that smaller cells might have proportionately fewer C-boutons than their larger counterparts. Since C-boutons are an excitatory input to motoneurons, it could make sense that smaller cells (which produce smaller, fatigue-resistant forces) would have proportionally less of these excitatory inputs than their larger-force counterparts. A larger study is required to test this hypothesis. In the meantime, we can conclude that the algorithm reliably measures changes in net protein expression and relative net expression consistently across different users of the software.

Additionally, both users consistently measured C-bouton size (2D LCA and 3D volume in Figure 7E,F, respectively), finding no significant differences between the three animals. These findings are promising, as they demonstrate that the algorithm enables comparable C-bouton measurements irrespective of the user performing the analysis, while still capturing biological variability across animals. Importantly, macro-cluster size remained consistent among the control animals. At present, we conclude that density may serve as a more sensitive indicator of relative net expression than relative total volume; however, this discrepancy warrants further investigation in a comprehensive study of C-boutons with a larger sample size.

### 3.3. Threshold Sensitivity Analysis and Selection Criteria

With validation and reproducibility testing completed for both soma size and protein expression, we next examined one of the most technically challenging aspects of our approach: thresholding. Thresholding is a critical step in analysis automation, serving as the algorithmic equivalent of visually distinguishing label from background—a major source of variability in manual analysis. Given its importance, we applied a data science framework to evaluate and optimize its impact on protein quantification and reproducibility.

To begin, we carefully visualized the identified proteins in both 3D and 2D, comparing them side-by-side with the raw labeled images across a range of threshold values. This allowed us to empirically determine thresholding boundaries—points where labeling was clearly too low (under-labeling) or too high (over-labeling). Using the thresholding algorithm and shift described in Section 2.5, we identified abundant over-labeling artifacts at a 5% threshold shift, which we designated as the lower bound of our threshold range. Conversely, we observed significant under-labeling at a 40% threshold shift, establishing this as the upper bound of our range. With these bounds defined, we systematically analyzed how our cluster measurements varied across the full 5–40% threshold range. Ultimately, we narrowed the range to 10–30%, as no cells had optimum thresholds outside of this range.

Figure 8 demonstrates high consistency in all cluster measures across the entire range of thresholds. Importantly, the measures that differed between animals in Figure 7 did not intersect at any point in Figure 8, indicating stable group differences regardless of threshold choice. Similarly, measures that were not statistically different in Figure 7 remained closely aligned in Figure 8, with minimal and visually steady separation across the threshold range. This consistency highlights the robustness of our relative comparisons, showing that threshold shifts do not significantly alter the observed patterns.

However, applying a single threshold shift across all cells introduced issues, including highly non-normally distributed cluster measures and outliers that were difficult to manage. These outliers typically resulted from cells that responded differently to thresholding due to inherent variability in factors such as label penetration, label specificity, random artifacts, and background signal level—common challenges in the immunohistochemistry process. To address this, we developed an objective and consistent way to adapt thresholds to account for this biological and experimental variability, thereby reducing thresholding-related outliers.
Figure 8Threshold sensitivity of macro-cluster measures for user 1. (**A**,**B**) Net expression. (**C**,**D**) Relative net expression. (**E**,**F**) Cluster Size. User 2 threshold sensitivity graphs looked nearly identical to user 1 and had no notable differences, so we chose to only display user 1’s results for clarity and simplicity. Each point represents the median of each animal at the indicated threshold shift. Gray shaded regions are the average steady threshold range for all cells, the midpoint of which was used for the auto-threshold applied throughout the results (each cell had its own range and midpoint, with midpoints ranging from 12.5% to 27.5%). Gray arrows indicate the average percent change in the parameter across the steady threshold range, and black arrows indicate the average percent change across the entire (10–30%) threshold range.
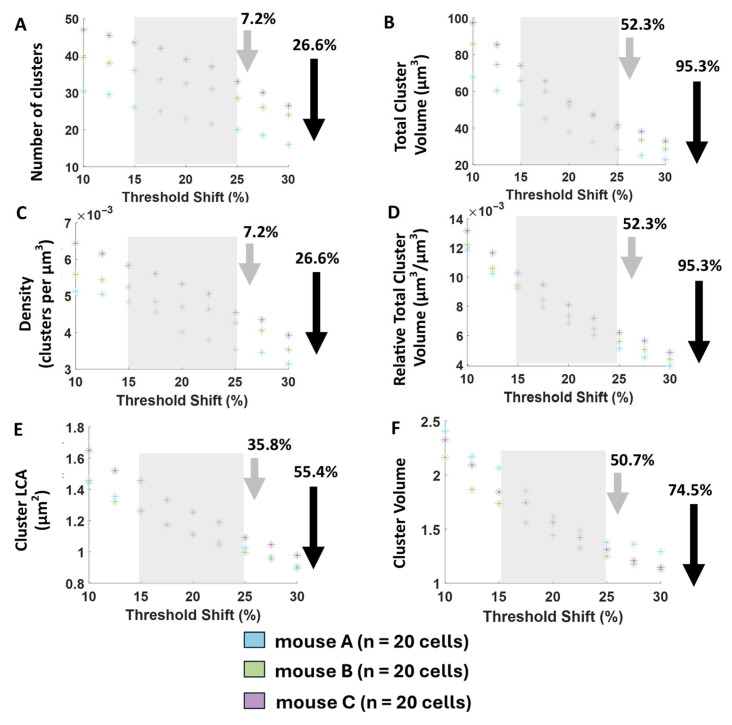



To address variability in background, artifact, and other image conditions across individual cells, we compiled detailed observations on how over- and under-labeling manifested within the cluster measurements. These insights informed us of the development of our automatic per-cell threshold adjustment algorithm. Specifically, we observed that cluster size and total volume often changed steadily across all threshold shifts, even at extreme over- and under-labeling conditions. Therefore, these parameters could not be used to differentiate a reasonable threshold range. Alternatively, the number of clusters was more promising for indicating threshold reasonability. We noted that for each cell, the number of clusters identified was more stable at thresholds that produced visually reasonable overlays, with larger deviations occurring at the poor overlays (obvious over- and under-labeling). We therefore used stability in the number of clusters to mathematically determine our reasonable threshold range for each cell (called the stable threshold range). The average of these cell-specific stable threshold ranges is illustrated by the gray shaded boxes in Figure 8. Gray arrows indicate the average % change in each parameter across the stability threshold range, while black arrows indicate the % change across the entire (10–30%) threshold range. We optimized cluster measurement stability by choosing the midpoint of each cell’s threshold stability range as its applied threshold. This cell-specific auto-thresholding approach was used for all algorithm-derived data presented in Figure 4 and Figure 7.

Although applying a uniform threshold shift across the entire dataset (as shown in Figure 8) yielded consistent trends in cluster measurements, our preliminary implementation of per-cell automatic thresholding provided an even greater benefit: complete consistency between users in the statistical conclusions drawn from hypothesis testing (Figure 7). The current performance of this automatic feature detection strongly indicates its potential for generating reliable and reproducible results.

As an illustration, Figure 9 shows one representative cell from each of the three control animals, displayed at extreme threshold shifts (5% and 40%), the default shift (10%), and the auto-suggested threshold.
Figure 9Three-dimensional visualizations at some key threshold shifts. One example cell per animal (each row) is displayed with algorithm-identified clusters at each relevant threshold shift (each column). At the 5% shift, we see over-expression and large clumps of clusters. At the 40% shift, we see bare patches of soma membrane and under-represented macro-clusters. Our 10% default shift appears reasonable but can lead to over-labeling of some cells (one example is the displayed cell of animal A). We found our auto-threshold shift for optimum cluster measure stability better handles these difficult-to-threshold cells without changing the trends of our data. Our auto-threshold enables our hypothesis statistics to be less manipulated by over-labeling that can unexpectedly occur on portions of some membranes due to random label artifact or other image conditions that are both tedious and difficult to identify visually and can be inconsistent from one user’s reconstruction to the next.
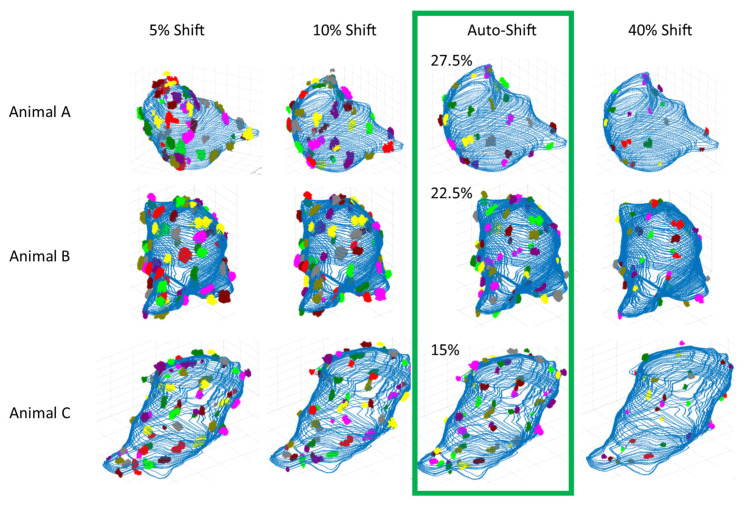



### 3.4. User Efficiency and Computation Metrics

Comparing user time per cell: The greatest efficiency improvement of our algorithm is in the manual user time per amount of data obtained. Manual analysis following the standard protocol described in Section 3.2.1 (tracing the cell LCA, tracing 5–8 en-fosse cluster LCAs, and performing density estimates) takes users, on average, 15 min per cell per protein of interest. This time can be as fast as 10 min per cell for an experience analyzer working with an ideal protein label. These reported times do not include the additional time to export and organize the measurements for analysis, which is automatic with our algorithm. On average, reconstructing the cells via our algorithm takes 10 min per cell. This can be as fast as 5 min per cell when the cell-body label is good and the user is experienced. Subsequent protein analysis with the algorithm is fully automated and thus does not add to the manual user time. Thus, we improve the manual time for analysis per cell by 5 min per cell, or 1/3 faster. Additionally, our algorithm measures more than six times the number of clusters per cell as manual methods (see Figure 1), making the total manual time/data obtained improved by a factor of 6/(2/3) = 9 for the protein cluster analysis.

Eliminating redundant analysis: After reconstruction with our algorithm, any number of proteins that were labeled (we label three proteins at a time) can be studied from the same set of reconstructions, whereas, with manual methods, the manual protein traces and density counts would need to be repeated for each protein. Likewise, with other 3D analysis software programs (e.g., Imaris or Neurolucida 360), the manual cell-by-cell tuning adjustments would need to be repeated for each protein. Furthermore, adjustments to analysis settings, like an alternative threshold approach or change in cluster definitions, are automatically iterated by our algorithm and so do not increase manual user time for algorithm analysis. In contrast, for manual or existing 3D software analysis, these changes would multiply the manual user time by the number of settings to be tested.

Appeal and per-day maximum: We also noted that manual analyzers all reported that one animal per day (20–30 cells) was their maximum “sanity” limit for manual analysis due to fatigue and restlessness. However, these same users could analyze two or more animals per day (with the same number of cells) using the algorithm, before fatiguing. One user reported completion of 3–4 animals per day with our algorithm. The ability of the user to not have to be in a dark room for any part of our algorithm workflow is a major factor that makes our analysis more appealing than methods that use visual tracing or cell-by-cell manual tuning.

#### Computation Metrics

Data storage: The formatted algorithm output of C-bouton measurements in Figure 7 and Figure 8 (at all thresholds and compiled auto-thresholds for easy stats) had the following size. Average of user 1 and user 2 excel workbook size: 36,941 KB or 617 KB per cell; and condensed matfile size: 3863 KB = 64 KB per cell.

Runtimes: This performance metric will vary based on the size of the dataset and the specifications of the computer on which the algorithm is run. We tested the computation time for the generation of the 60 cell reconstructions used in Figure 7 and Figure 8 and their protein analyses on two separate machines. Our Windows machine has 128 GB of random-access memory (RAM) and 12 cores. Our MacBook Air (M2 chip) has 6 GB of RAM and 8 cores.

The reconstructions took 34.38 s/cell to compute on our Windows machine and 27.23 s/cell on our MacBook Air. Measuring the reconstructed somas took the algorithm 1.31 s/cell on the Windows machine and 1.27 s/cell on the MacBook Air. For the automated protein cluster analysis across all thresholds, we ran the algorithm in parallel on nine cores (one threshold per core) on our Windows machine. The time to generate all cluster data and save to the excel workbook was 81.47 min, or 1.36 min/cell. The auto-threshold per cell feature took an additional 70 s, or 1.17 s/cell. Due to the parallel processing, these times represent how long it took the longest (lowest) threshold to complete. Higher thresholds take less time to compute. Therefore, if multiple cores are not available or RAM is limited, the analysis can be run serially for all nine thresholds, with a total run time less than nine times those reported for parallel processing. For example, on our MacBook Air, due to the limited application memory, we ran the cluster analysis serially with a total time to generate all cluster data and save to the Excel workbook of 231.08 min or 3.85 min/cell. The auto-threshold-per-cell feature took an additional 78 s or 1.30 s/cell. We found these runtimes and memory requirements acceptable for our analyses, but we do expect even further improvements through ongoing computational optimization.

## 4. Discussion

Characterizing soma size and protein expression patterns is critical for linking motoneuron structure to function under both normal and pathological conditions. This is especially important in neurodegenerative research, such as studies of aging and motoneuron disease, where understanding structural changes in somas and their associated proteins under altered electrical activity may reveal mechanisms of degeneration and identify potential targets for therapeutic intervention aimed at preserving motoneuron function and preventing or slowing down cell death.

Immunohistochemistry (IHC) is a powerful tool that enables the visualization of motoneuron somas and associated proteins for structural analysis. However, despite its strength, IHC is often underutilized as a source of primary quantitative data due to challenges in objectively interpreting and quantifying protein expression, as IHC images often exhibit high variability in label intensity and background noise. These inconsistencies make fair and consistent thresholding across images particularly difficult. Furthermore, existing analysis tools are primarily designed for analysis of brain tissue and typically require extensive manual effort to identify and measure protein clusters, leading to subjective sampling, limited throughput, and poor reproducibility. These issues contribute to inconsistent findings between studies—and even within the same dataset analyzed by the same individual at different times [2].

To address these limitations, we developed a custom software tool, independent of costly commercial platforms, specially designed to facilitate consistent comparisons of soma size and protein expression across IHC images in spinal tissue. Importantly, our automated analysis is inherently blind to study groups and is order-agnostic, meaning the results are independent of the order in which the cells are analyzed (removing the need for order randomization).

Soma size: Our Cartesian-based approach enables precise 3D measurement of soma size at the highest possible voxel resolution. This represents a significant improvement over many commercial tools, which often rely on adaptive cylindrical compartment models with varying sizes and, consequently, inconsistent resolution. In contrast to these tools—which do not disclose the accuracy of their volume or surface area calculations—we transparently report a consistent 23% underestimation in volume. This known and reproducible bias allows for accurate comparisons of soma size across experimental groups and offers the option to apply a correction factor to estimate true physiological soma volume.

Importantly, our results show that volume is the most appropriate and reliable metric for assessing soma size. We also show that surface area is not a reliable measure at this resolution, due to its inability to accurately capture complex 3D geometry. Our automated edge detection method enhances the efficiency, objectivity, and physiological relevance of both 2D and 3D soma reconstructions. Additionally, we confirm that 3D volume is more sensitive than 2D LCA in detecting subtle changes in soma size.

Protein expression: We developed an automated approach to replace the traditionally tedious and subjective manual analysis of protein expression, significantly improving both objectivity and consistency. This automation also greatly enhances analytical efficiency by eliminating the need for manual protein cluster sampling and tracing, making it feasible to process large datasets with high cell counts in a fraction of the time previously required.

Furthermore, all algorithm settings are user-accessible numerical values, allowing for transparent control and easy reporting—critical for ensuring reproducibility. For the first time, our algorithm enables efficient and systematic exploration of various threshold and cluster definition scenarios, allowing us to track measurement sensitivity and verify the robustness of our results.

With this new level of reliability and rigor, IHC-based protein expression data can now serve as primary quantitative results in studies—yielding reproducible, biologically meaningful insights that advance our mechanistic understanding of motoneuron function.

Limitations and other insights: One major limitation of this analysis is the necessary assumption that the tissue, images, and label are all in fair condition and reasonably represent the physiology being studied. We learned, from our extensive testing during development, that directly comparing protein expressions from images taken in separate experiments, antibody batches, or under different protocols could easily confound and confuse the conclusions of a study. Notably, these limitations also exist in prior analysis methods, but are often “compensated for” by manual adjustments or corrections during analysis. Although these manual corrections are fairly common practice, these adjustments introduce great bias and subjectivity, and are likely a large contributor to the lack of rigor and reproducibility that plague the field. Our algorithm should be used to analyze images from tightly controlled experimental conditions, where all groups in a study are processed in batches, i.e., each batch of processing contains tissue from all study groups. The algorithm does not perform corrections by filtering or adjusting brightness levels to meet visual assumptions of how the label is expected to look. Rather, our auto-threshold adjustments and DBSCAN parameters are applied objectively, mathematically, traceably, and consistently to all cells in the study.

Another limitation to consider is image resolution. For this validation study on motoneuron somas and somatic C-boutons, our sampling resolutions were 0.172 microns/pixel and 0.3 microns/slice. It should be noted that our reported 23% volume underestimation was based on this resolution, and could vary at alternative resolutions. We would expect a smaller underestimation for higher resolutions, and a larger underestimation for lower resolutions. For analyzing motoneuron somatic protein structure, we do not recommend lower resolutions given the small size of the proteins.

Future work: In this study, our analysis of protein expression focused on the C-bouton, a structure known to form distinct macro-clusters on the motoneuron somatic membrane. However, our algorithm is also capable of analyzing proteins that exhibit more diffuse or less distinct clustering patterns, including those with variable levels of phosphorylation. Determining the most effective methods to quantify these proteins—both in terms of phosphorylation state and total expression—across different conditions is a key focus of our ongoing research. Additionally, our algorithm is also equipped to study proteins in other key regions of spinal or brain tissue, either isolated to cell geometry areas or over entire image regions. We are continuing to expand the scope through additional features and workflows that can be customized for a variety of applications.

Broader implications and perspectives: Although this study was focused on spinal tissue and motoneuron analysis, accurate segmentation and quantification remain significant challenges in other basic research and clinical diagnostic applications involving immunohistochemical imaging [10,11]. Our algorithm improves on several subjectivity, usability, and reproducibility issues that plague alternative analysis algorithms and graphical user interface (GUI)-based qualitative approaches. Our specialized level of automation and rigorous stability testing are among the features that make our algorithm unique. Alternatively, while not yet utilized for motoneuron analysis, deep learning is commonly leveraged in other IHC applications to perform desired image segmentations. While conceptually promising, these deep learning techniques have proven to have limited accuracy and generalizability so far, with no clear model consistently outperforming other models across datasets [10,11]. Future studies and generalizations of our algorithm are needed to compare performance to such State-of-the-Art (SOTA) deep learning models outside the context of motoneuron analysis.

## 5. Conclusions

We present and validate a novel, rigorous approach for IHC analysis, specifically designed to quantify soma size and somatic protein expression in motoneurons. While tailored to motoneuron analysis, the algorithm is flexible and can be adapted for use with other neuron types and protein markers. We aim for this tool to serve as an approachable, efficient, and standardized solution for IHC analysis—helping to resolve inconsistencies and improve the reliability of quantified data across the field.

## 6. Patents

The authors disclose that the technology described in this manuscript has been disclosed to Wright State University’s Technology Transfer Office, and a provisional patent application will be filed before publication.

## Figures and Tables

**Figure 1 bioengineering-12-00761-f001:**
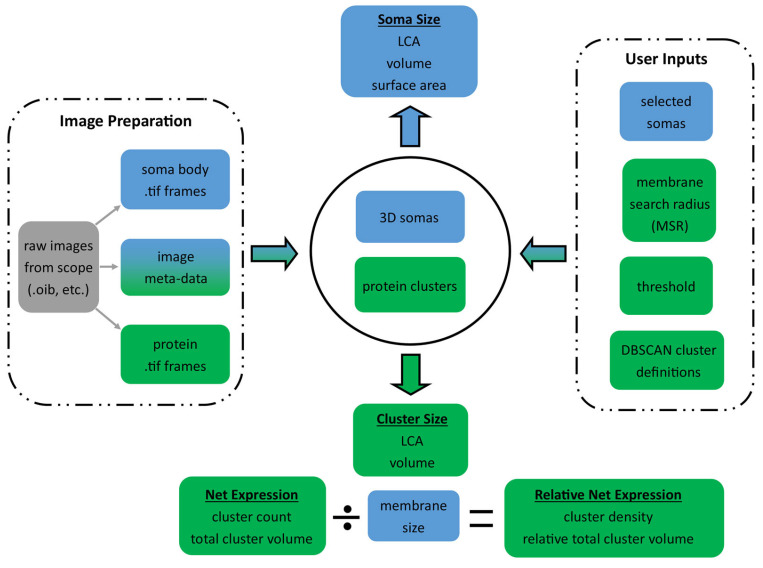
Algorithm box diagram. The image files on the left and user inputs on the right are fed into the central algorithm. Parameters associated with the 3D soma reconstructions are indicated by blue backgrounds, and parameters associated with the protein clusters are indicated by green backgrounds. The algorithm quantitates the auto-identified soma and protein cluster structures and then outputs the soma measures and protein cluster measures. These outputs can be compared across multiple combinations of user inputs by simply looping the algorithm through different scenarios. All data is organized and saved to Microsoft Excel workbooks for easy plotting and hypothesis testing.

**Figure 2 bioengineering-12-00761-f002:**
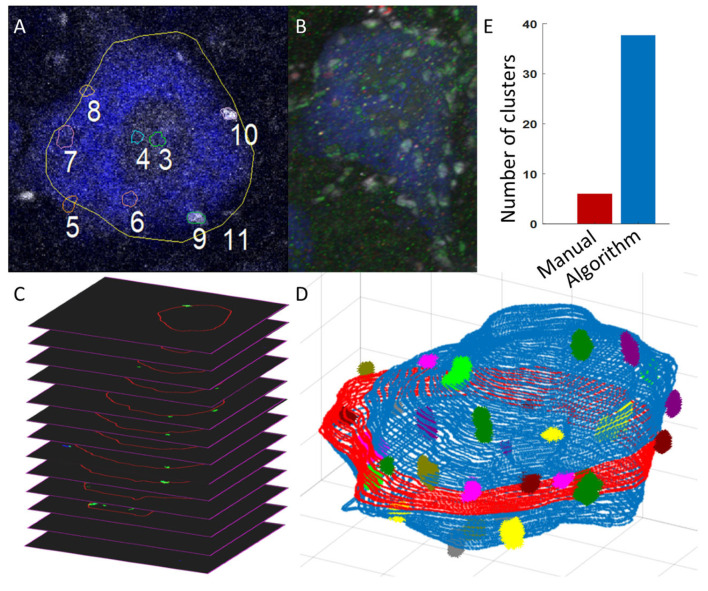
Visualization of one cell and sampling comparisons between manual and algorithm methods. Manual methods with existing software, showing (**A**) one manually traced blue soma (trace 11, measuring the 2D largest cross-ectional area (LCA)) and 8 smaller manually traced en-fosse white C-bouton protein clusters (traces 3–10) in Fluoview; and (**B**) 3D rendering of the blue soma and white C-boutons in Neurolucida. (**C**) Stack of our 2D auto-generated soma outline (red ROIs) and identified C-bouton protein label displayed in green. (**D**) Our 3D visualizer with auto-generated soma reconstruction and auto-identified protein clusters. The red portion in the middle of the cell indicates the slices on which manual cluster counts would be measured to approximate cluster density. (**E**) The average number of clusters measured per cell for cluster size comparisons in manual vs. algorithm methods. Manual average was taken from Dukkipati et al., 2016 study [2] from 38 control cells. Algorithm average was from 60 control cells (average of two users’ measurements for each cell).

**Figure 3 bioengineering-12-00761-f003:**
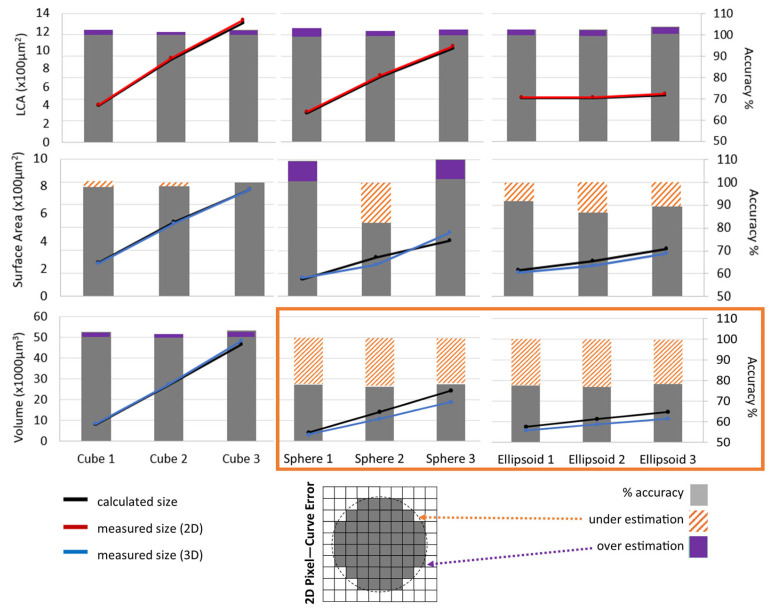
Evaluating algorithm accuracy across different geometric shapes. LCA, surface area, and volume of known 3D shapes of 3 different sizes (small, medium, and large). The left axes indicate the sizes, graphed as lines, with black for mathematically calculated, and red or blue for 2D or 3D algorithm measurements, respectively. The right axes indicate percent accuracy of the measurements compared to the calculated sizes, displayed as gray bars. Purple overlay indicates the algorithm measured larger (overestimation), while the orange striped filling indicates the algorithm measured smaller (underestimation). The orange box emphasizes the consistent volume under estimations for the curved geometries. The bottom panel between the legends is a 2D visual example of a pixel–curvature error, which becomes a voxel–curvature error in 3D. White areas within the dashed circle add to underestimation, and gray areas outside the circle add to overestimation.

**Figure 4 bioengineering-12-00761-f004:**
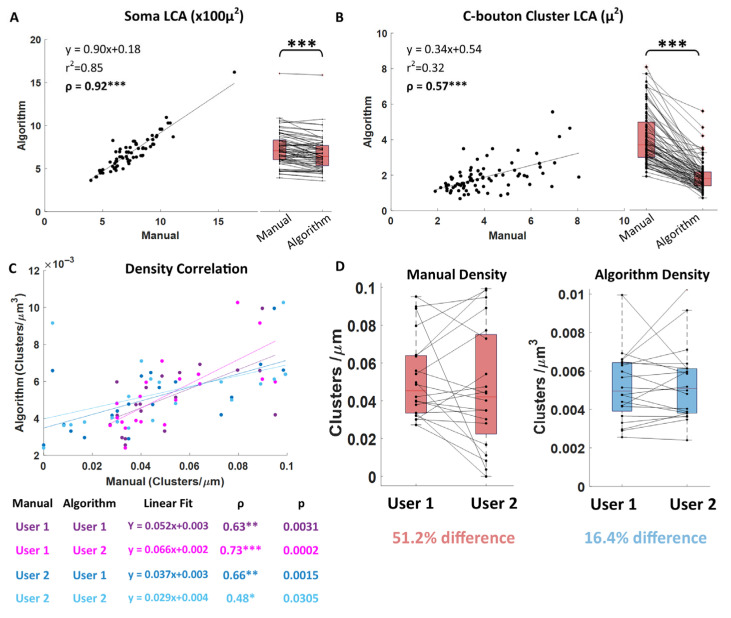
Algorithm comparison to manual measurements. (**A**) Algorithm soma LCA correlates to manual measurements (*p* < 0.0001) and measures 8.2% smaller than the manual traces (paired Wilcoxon, *p* < 0.0001). (**B**) Algorithm C-bouton cluster LCA correlates to manual measurements (*p* = 0.0017) and measures 72.4% smaller than the manual traces (paired Wilcoxon, *p* < 0.0001). (**C**) The 3D algorithm density using 100% of the soma membrane correlates to the manual density approximation from 25% of the membrane for the same 20 cells, but the strength of this correlation varies depending mostly on the manual user (notice that the linear regression lines form 2 groups of blue and pink). (**D**) Manual and algorithm densities compared pairwise for the same 20 cells between the 2 users. The manual density measurements vary between users by an average of 51.2%, while the algorithm density measurements vary between users by an average of 16.4% (*p* < 0.05 *, *p* < 0.01 **, and *p* < 0.001 ***).

**Figure 5 bioengineering-12-00761-f005:**
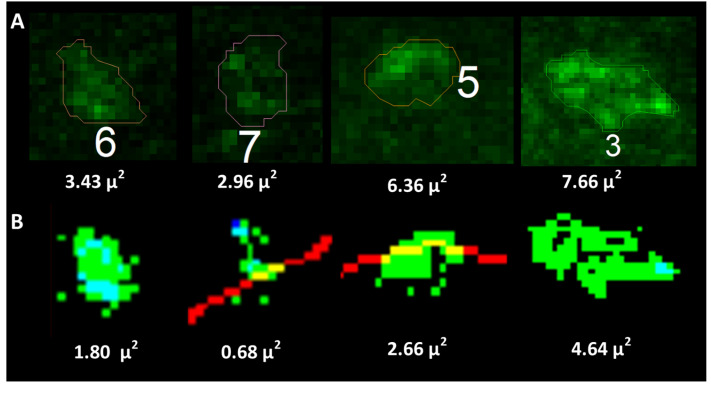
Visualizing algorithm cluster LCAs vs. manual traces. (**A**) Manual traces of 4 example clusters (traces 6, 7, 5, and 3), with the manual LCA in white text under the cluster. Notice the tendency toward an elliptical shape in all (but especially the first 3) clusters. Also, notice some very dim pixels near the inside cluster edges, suggesting these clusters were traced with an increased brightness for visualization and are thus overestimated. There are also some dim pixels within the centers of the clusters, which would be included as part of the labeled area in the manual LCA. (**B**) The same clusters overlayed with bright green as identified and measured by the algorithm at the 10% default threshold shift. The red lines are nearby soma edges. In this visualizer, the original C-bouton label intensity is displayed in dark blue. Dark blue indicates pixels that had visual label but were not included as part of the cluster because they (1) were too far from the soma membrane, (2) were not bright enough to meet the threshold for label, or (3) did not meet the criteria to be a part of that cluster. The single dark blue pixel in these clusters (specifically in the second from the left cluster) is illustrative of the rarity of any of these 3 scenarios since the membrane search radius, threshold shift range, and cluster criteria were tuned during algorithm development to reduce the occurrence of dark blue pixels. The light blue/teal pixels had bright enough raw blue intensity to shine through the green “identified label” overlay and are part of the identified clusters.

**Figure 6 bioengineering-12-00761-f006:**
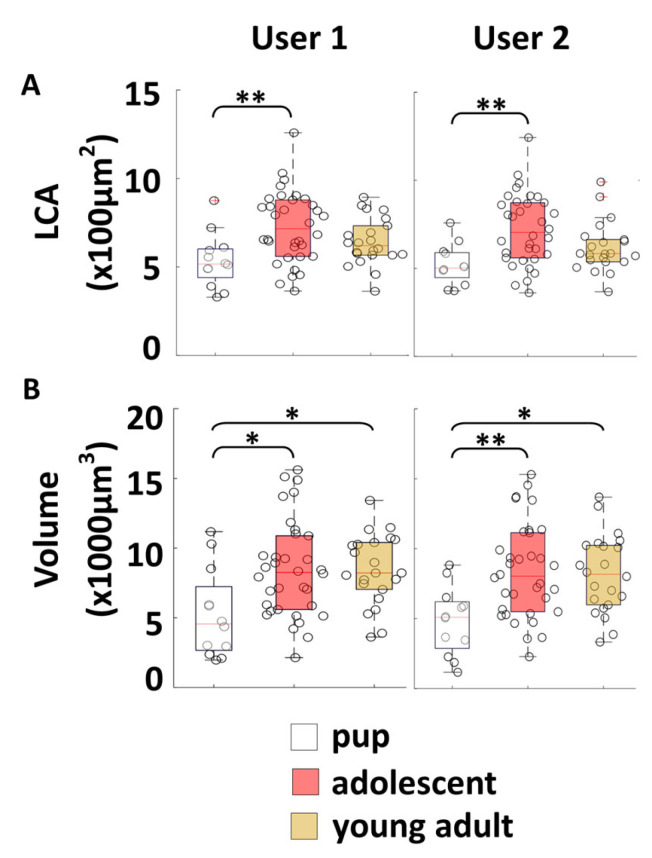
Reproducibility and specificity of algorithm soma measurements. (**A**) LCA and (**B**) volume measured by two different users of the algorithm, analyzing their separate reconstructions of the same cells (*p* < 0.05 *, *p* < 0.01 **).

**Figure 7 bioengineering-12-00761-f007:**
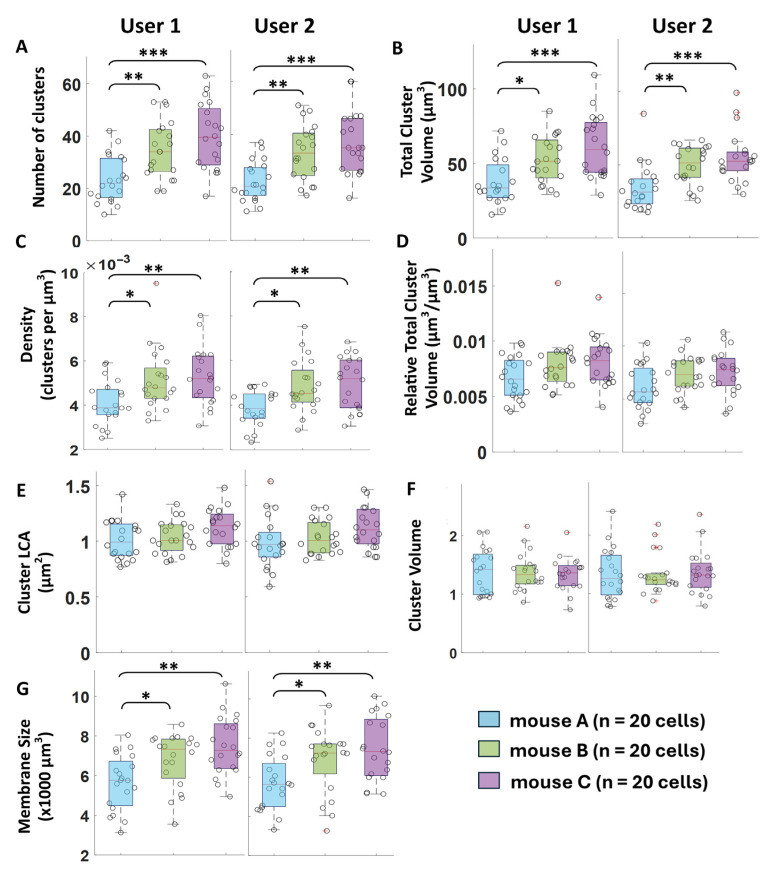
Reproducibility of algorithm macro-cluster measurements. All measures were gathered independently by 2 separate users, with their own cell reconstructions. (**A**) Number of clusters, (**B**) total cluster volume, and (**C**) cluster density (# clusters per soma membrane size) were less in animal A than animals B and C. (**D**) Relative total cluster volume (total cluster volume per soma membrane size) was not different between the 3 control animals. The cluster size ((**E**) 2D LCA and (**F**) 3D volume) was not different between the 3 control animals. (**G**) Soma membrane size was smaller in animal A than animals B and C, underlying part of the net expression differences in subfigures (**A**,**B**) (*p* < 0.05 *, *p* < 0.01 **, and *p* < 0.001 ***).

## Data Availability

The original contributions presented in this study are included in the article. Further inquiries can be directed toward the corresponding authors.

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
