# Peer review of "Precision in 3D: A Fast and Accurate Algorithm for Reproducible Motoneuron Structure and Protein Expression Analysis"

_bioengineering, 2025, doi:10.3390/bioengineering12070761_

Round 1
Reviewer 1 Report
Comments and Suggestions for Authors
- On page 3, line 13, the term "Cartesian" is used in place of "Cartesian." On page 14, line 40.1, the reference "Figure 6A-B" should be replaced with the reference "Figure 7A-B".
- Please refer to the following caption on page 3, figure 1: It would be beneficial to provide an explanation of "DBSCAN cluster definitions" at the point of its initial mention. DBSCAN, an acronym for "Density-Based Spatial Clustering of Applications with Noise," is a specific type of clustering algorithm.
- Systematic volume error (23%). While the authors accurately identify and quantify a systematic underestimation of volume by approximately 23% for curved objects (see Figure 3), they do not provide a convincing explanation of the physical or algorithmic cause of this error. The assertion that the phenomenon under discussion is associated with "the level of curvature and dimensional resolution" (p. 6) is overly ambiguous.
4. Thresholding analysis. As delineated in Section 2.3.1, the selected thresholding method—that is, the triangle algorithm with a 10% shift—along with its rationale, i.e., visual satisfaction, is examined, as is its sensitivity analysis. It is imperative to ascertain the specific, objective criteria that were utilized to determine the "stability" of cluster measurements across the threshold range. The quantification of "over-labeling" and "under-labeling" at the cellular level was essential for constructing the stability range.
The sensitivity analysis (Figure 8) demonstrates the stability of relative differences between groups; however, it does not illustrate the extent to which the absolute values of the measures (e.g., number of clusters, total volume) are altered in absolute terms when the threshold is modified. The extent of this variability is a critical question that merits rigorous investigation.
- Relative cluster volume vs. cluster density (Figure 7C, D): The results show significant differences in cluster density between animals A and B/C, but not in relative total cluster volume. The authors suggest that density may be a more sensitive measure. However, it is unclear why these two measures, both claiming to reflect relative net expression (normalized to membrane size), yielded different results. Cluster volume depends on both the number of clusters and their size. Cluster size did not differ (Figure 7E, F). Does this mean that the difference in Density is due solely to the difference in the number of clusters? If so, why did the Relative Total Volume (which = Density * Average Cluster Volume) show no differences if the Average Cluster Volume is constant? This seems mathematically contradictory unless the average cluster size compensated for the difference in number (e.g., animal A has fewer clusters, but they are larger). However, Figure 7E, F does not show differences in size.
- The key advantage claimed is increased efficiency. However, there is no data quantitatively comparing the analysis time per cell/dataset between the new algorithm (including user reconstruction time) and manual/semi-automatic methods.
Reviewer 2 Report
Comments and Suggestions for Authors
1. The validation and reproducibility tests were conducted using images collected in-house over the last decade. However, the robustness and generalizability of the algorithm would be strengthened by testing it on external datasets from independent laboratories with varying imaging conditions and labeling practices.
2. The manuscript strongly emphasizes the advantages of the proposed algorithm, but does not critically discuss its limitations, such as how it performs with damaged tissue, potential failure cases in noisy or low-resolution images, and difficulty with proteins not located at the membrane.
3. While the algorithm's performance is well-validated, the biological interpretation of the C-bouton protein expression differences is lacking. For example, what are the physiological or pathological implications of reduced C-bouton density in smaller motoneurons?
4. There is no direct comparison of runtime, memory usage, or segmentation accuracy against commercial software in quantitative terms. Without this, it is hard to assess how much more efficient or accurate the algorithm really is.
5. The algorithm’s validation heavily relies on visual overlays and subjective inspection of threshold stability (e.g., Figure 9). While helpful, more rigorous quantitative metrics such as Dice coefficient or Jaccard index would support claims of segmentation accuracy.
6. In the manual vs. algorithm comparisons, the algorithm often performs better, but the manual traces were performed under conditions that may unintentionally bias against them (e.g., increased brightness artificially inflating boundaries). This should be clearly acknowledged and controlled for in fairness.
7. The manuscript lacks a statement regarding the availability of the algorithm code or GUI, which is essential for reproducibility by other researchers and broader adoption. Open-source availability would significantly enhance impact.
Round 2
Reviewer 2 Report
Comments and Suggestions for Authors
- There are only nine references in this research article, with no references from recent years.
- The authors are suggested to read and add the recent literature to support their findings and results.
- Under the results discussion, a suggestion is to compare the findings of literature studies to showcase the novelty of the reported work.
- There are grammatical typos in sentence making throughout the manuscript. Please double-check.
